A cross-sectional study on the effects of bedtime administration of selective α1 adrenoceptor antagonists on nocturnal blood pressure in elderly patients with benign prostate hyperplasia

Chen Chao-Ting 1
Ma Shao-Jun 1
Wang Hai-Ya Wanghy2024@yeah.net 1
Yao Hai-Jun 2
1 Department of Geriatrics, Shanghai Ninth People’s Hospital, Shanghai Jiaotong University School of Medicine , Shanghai , China
2 Department of Urology, Shanghai Ninth People’s Hospital, Shanghai Jiaotong University School of Medicine , Shanghai , China
Barbosa Neto Octavio
Electronic publication date: 2025 Apr 1
Publication date: 2025
Volume: 13
Electronic Location ID: e19165
Received 2024 Jul 3; Accepted 2025 Feb 24
Copyright: ©2025 Chen et al.
Copyright year: 2025
Copyright holder: Chen et al.
License: This is an open access article distributed under the terms of the Creative Commons Attribution License, which permits unrestricted use, distribution, reproduction and adaptation in any medium and for any purpose provided that it is properly attributed. For attribution, the original author(s), title, publication source (PeerJ) and either DOI or URL of the article must be cited.
License URL: https://creativecommons.org/licenses/by/4.0/

Keywords: Nocturnal hypotension, Ambulatory blood pressure monitoring, Adrenergic alpha-1 receptor antagonist, Geriatric medicine, The elderly

Funding: Shanghai Municipal Health Commission Foundation 202040076 202140030 Geriatrics Department Development Fund of Shanghai Ninth People’s Hospital, Shanghai Jiao Tong University School of Medicine JYXKA202012 This study was supported by the Shanghai Municipal Health Commission Foundation (202040076 and 202140030) and the Geriatrics Department Development Fund of Shanghai Ninth People’s Hospital, Shanghai Jiao Tong University School of Medicine (JYXKA202012). The funders had no role in study design, data collection and analysis, decision to publish, or preparation of the manuscript.

==============================
Background

It remains uncertain whether a bedtime dose of selective α1 adrenoceptor antagonist could result in nocturnal hypotension in elderly patients with benign prostate hyperplasia (BPH).

Methods

A total of 253 older men with BPH who had taken selective α1 adrenoceptor antagonists before sleep were consecutively recruited from the Geriatric Department of Shanghai Ninth People’s Hospital. A total of 221 patients were finally included in the analysis with qualified data including office blood pressure examinations, biochemical tests of blood, and 24-hour ambulatory blood pressure monitoring. Nocturnal hypotension was defined according to the nighttime average systolic blood pressure of ambulatory blood pressure ≤ 100 mmHg and/or diastolic blood pressure ≤ 60 mmHg. Explore the presence of night hypotension, compare the characteristics of the two groups with or without nocturnal hypotension, and analyze the related risk factors.

Results

Among all 221 patients included in the analysis, nocturnal hypotension occurred in 38 patients (17.2%). Compared with those without, patients with nocturnal hypotension were older, had less body mass index, lower office diastolic blood pressure, and lower ambulatory blood pressure in a 24 hour day, and night systolic and diastolic blood pressure, and were less likely to have hypertension. Age (OR 1.064, 95% CI [1.012–1.118], P = 0.015) and no hypertension (OR 2.548, 95% CI [1.211–5.359], P = 0.014) were independently associated with the presence of nocturnal hypotension.

Discussion

Nocturnal hypotension was common in men 60 years and older with BPH treated with selective α1 adrenoceptor antagonists before sleep. Age and no hypertension were independently associated with nocturnal hypotension positively. Related factors may help clinicians identify hypotension tendencies in the elderly when prescribing such drugs.

Introduction

Selective α1 receptor antagonists (sα1-RAs) are widely prescribed for relieving lower urinary tract symptoms (LUTS) caused by benign prostatic hyperplasia (BPH), a most frequent disease in aging men (Lowe, 2004). They are often taken before sleep as doctors’ recommendation to avoid the risk of postural hypotension, especially in elderly patients, though uroselective, and have fewer adverse effects on systemic blood pressure (BP) compared to non-sα1-RAs (Li et al., 2022; Madersbacher, Sampson & Culig, 2019). Increasing evidence (Kario et al., 2001; Metoki et al., 2006; Pierdomenico et al., 1998) suggested that excessive reduction of BP during sleep was associated with cardiovascular events in elderly hypertensives or members of the general population, deserving attention just like the relationship of morning BP with cardiovascular outcomes and vascular damage (Madin & Iqbal, 2006; Li et al., 2010; Chen et al., 2014). Nocturnal hypotension (NHP), conferring the risk of organ perfusion, was considered the critical cause of unfavorable cardiovascular events (Pierdomenico et al., 1998; Kario et al., 1996) in patients even with hypertension. Usually, the normal circadian rhythm of blood pressure is high during the day and low at night. As we know, α1-RAs may have effects on systemic blood pressure. It is unclear whether a nighttime dosage of sα1-RAs during the dose stabilization period after drug titration will exacerbate the decrease in nighttime BP and alter the original dipping pattern. Would the currency of nocturnal hypotension increase particularly in elderly individuals without hypertension and those with a circadian rhythm of extreme dippers? The current cross-sectional study investigated the prevalence and related risk factors of nocturnal hypotension based on ambulatory blood pressure monitoring (ABPM), paying attention to the growing elderly population with the sα1-RAs treatment for benign prostatic hyperplasia (BPH) easily neglected in clinical practice.

Materials & Methods

Study population

This study was conducted within an ongoing prospective study on the prognosis of abnormal nocturnal blood pressure in elderly patients approved by the Ethics Committee of Shanghai Ninth People’s Hospital (SH9H-2020-T25-2). According to the selection and exclusion criteria, a total of 253 elderly male patients with a history of BPH were consecutively recruited from the outpatient department and ward of the Geriatric Department at Shanghai Ninth People’s Hospital, from April 2020 to July 2021. Inclusion criteria are as follows: male patients, age ≥ 60 years; with BPH diagnosed by B-ultrasound and treated with a bedtime dose of sα1-RAs (including tamsulosin 0.2 mg, doxazosin 4 mg, and terazosin 2 mg) for more than three months. Exclusion criteria included diverse types of tumors, acute or chronic heart failure, uremia, Parkinson’s disease, acute cerebrovascular accident, or sequelae leading to long-term bedridden patients and nasal feeding or poor nutritional status with body mass index (BMI) below 18.5 kg/m2. All participants signed informed consent.

Ambulatory BP measurement

BPs were measured every 20 minutes during the day and every 30 minutes at night by 24-hour ABPM devices with the automated system using electrical cuff inflation (TM-2430; A&D Co., Tokyo, Japan). The qualified ABPM standard referred to our previous study (Chen et al., 2014) (at least one BP reading per hour for 24 h, monitoring times with adequate BP ≥ 70%, daytime BP reading ≥ 20, nocturnal BP readings ≥ 7). Nocturnal hypotension was defined according to the night-time average systolic blood pressure (SBP) of ABPM ≤ 100 mmHg and, or diastolic blood pressure (DBP) ≤ 60 mmHg mainly considering the predictive value of adverse cardiovascular events (Kikuya et al., 2007; Ungar et al., 2009; Protogerou et al., 2007). We subclassified circadian rhythm into normal (dippers) and abnormal (including non-dippers, reverse dippers, and extreme dippers). The dipping patterns of ABPM were referred to Kario, Schwartz & Pickering (2000)’s study (extreme-dippers, with ≥ 20% nocturnal BP fall; dippers, with ≥ 10% but < 20% fall; nondippers, with ≥ 0% but < 10% fall; and reverse-dippers, with < 0% fall).

Other measurements

We obtained clinical and laboratory variables through a standardized questionnaire. It collected information on inpatient and outpatient care, including demographic information, medical history, office BP, ABPM, BMI, fasting glucose, total serum cholesterol, and information on lifestyles such as current smoking and current use of tobacco. Office BPs were obtained by an automated machine (HBP-1300; OMRON Co., China) in patients who had rested for ≥5 min in the sitting position. Two consecutive BPs were measured with a 1-minute interval at the nondominant arm, and the average of the two BP values was recorded. Body height and weight were measured without shoes and wearing light indoor clothing for the calculation of body mass index (weight (kg)/height (m2)). Venous blood samples were collected after fasting for at least 8 hours and analyzed for glucose and serum total cholesterol by a Beckman Coulter AU 680 (Brea, CA, USA). Hypertension was defined by a self-reported diagnosis or a mean conventional systolic BP ≥ 140 mmHg or diastolic BP ≥ 90 mm Hg on ≥3 occasions or antihypertensive medication. Diabetes mellitus was defined by a self-reported diagnosis or as a fasting blood glucose ≥ 7 mmol/L or a postprandial 2-hour blood glucose ≥ 11.1 mmol/L or glycated hemoglobin A1 ≥ 6.5% or use of antidiabetic drugs. All patients with diabetes mellitus were type 2 not type 1.

Research group

After informed consent, all patients completed office BP measurements, relevant biochemical tests, and ABPM. After summarizing the 24-hour ABPM and biochemical test reports, eliminating 15 cases with invalid ABPM, and excluding 17 patients with unchecked or incomplete biochemical tests, 221 patients were included in the analysis. Two groups were divided in line with the above definition: NHP and no NHP.

Statistical analysis

We use the Statistical Package for the Social Sciences (SPSS) version 24.0 software (IBM Corp., Armonk, NY, USA) for database management and statistical analyses. Measurement data were expressed as mean ± standard deviation and means between the two groups were compared by two independent sample t-tests or rank-sum tests. Count data were expressed by percentage and analyzed by the Chi-Square or Fisher’s test. Univariate and multiple logistic regression analyses identified the risk factors associated with nocturnal hypotension. In the chi-square test of the distribution of nocturnal hypotension in different age groups, we used the SAS version 9.3 software (SAS Institute Inc., Cary, NC, USA) to perform the Cochran-Armitage trend test. In the multivariate adjustment analysis and multivariate joint prediction ROC curve, covariates were included as age, length of taking sα1-RAs, body mass index, total cholesterol, fasting blood glucose, eGFR, office SBP and DBP, current smoking, current drinking, diabetes, hypertension, sα1-RAs types and take other antihypertensive drugs at night. According to the size of the area under the ROC curve (AUC), receiver operating characteristic (ROC) analyses were used to summarize the diagnostic power of the related risk factors for discrimination between NHP and no NHP patients. Optimal cut-offs were derived from the ROC curves by maximizing the sum of sensitivity and specificity. A two-sided test, P < 0.05, was considered statistically significant.

Results

Participant characteristics

Among 252 recruited male patients, 221 eligible patients (mean age 69.1 ± 7.1 years, minimum 60, maximum 87) agreed to participate in the study, with an average duration of taking sα1-RAs 2.7 ± 0.8 years and an average body mass index (25.8 ± 2.9 kg/m2, minimum 19.1, maximum 36.7). There were 151 patients with hypertension, 44 patients with diabetes, 61 cases of current smokers, and 32 cases of current drinkers among all patients. In 38 cases taking other antihypertensive drugs at night, 6(15.8%) patients had nocturnal hypotension and 32(17.5%) without. There were 57 cases with the normal circadian rhythm of the dipper type, and 164 cases with abnormal circadian rhythm types, including 108 cases of non-dipper, 30 cases of reverse dipper, and 26 cases of extreme dipper. The mean office SBP and DBP were 150.3 ± 21.4 mmHg and 88.8 ± 11.8 mmHg, respectively. Of the 221 participants, 38 patients (17.2%) had nocturnal hypotension according to night-time average SBP and/or DBP of ABPM. Among those 38 patients with nocturnal hypotension, 9 (23.7%) had only low SBP, 12 (31.6%) had only low DBP, and 17(44.7%) both by ABPM.

Table 1 Clinical characteristics according to the presence or absence of nocturnal hypotension in 221 elderly patients with benign prostate hyperplasia.

Characteristic	No NHP	NHP	P Value	
Total, n (%)	183	38		
Hypertension	131 (71.6)	20 (52.6)	0.022*	
Diabetes mellitus	39 (21.3)	5 (13.2)	0.252	
Current drinkers	27 (14.8)	5 (13.2)	0.799	
Current smokers	52 (28.4)	9 (23.7)	0.553	
Take other antihypertensive drugs at night	32 (17.5)	6 (15.8)	0.801	
sα1-RAs types				
Tamsulosin	85 (46.4)	11 (28.9)	0.046*	
Doxazosin	58 (31.7)	12 (31.6)		
Terazosin	40 (21.9)	13 (39.5)		
Circadian rhythm				
Normal	43 (23.5)	14 (36.8)	0.087	
Abnormal	140 (76.5)	24 (63.2)		
Age, years	68.7 ± 7.0	71.4 ± 7.1	0.048*	
Body Mass Index, kg/m2	26.0 ± 2.8	24.8 ± 3.1	0.021*	
Fasting blood glucose, mmol/L	5.3 (4.8–5.8)	5.1 (4.6–5.6)	0.689	
Serum total cholesterol, mmol/L	4.5 ± 0.8	4.6 ± 1.0	0.404	
eGFR, ml/min/1.73 m2	0.3 (0.3–0.4)	0.4 (0.3–0.4)	0.172	
Duration of medication, years	2.7 ± 0.8	2.6 ± 0.9	0.354	
Office SBP, mm Hg	151.0 ± 20.8	149.2 ± 24.7	0.726	
Office DBP, mm Hg	90 (80–98)	80(76–90)	0.006*	
Ambulatory Blood Pressure, mm Hg				
SBP24 h	132.5 ± 14.4	112.6 ± 9.9	0.000†	
DBP24 h	81.4 ± 7.7	67.3 ± 6.2	0.000†	
SBPd	135.2 ± 15.8	120.8 ± 12.2	0.000†	
DBPd	83.3 ± 9.3	72.4 ± 7.4	0.000†	
SBPn	125.6 ± 17.3	98.1 ± 10.5	0.000†	
DBPn	76.5 ± 8.7	58.1 ± 5.8	0.000†	
Notes.

Data are the number (% of the column) of patients, mean ± SD, or median (interquartile range). P Values are for a 2-group comparison of means (student t-test or rank-sum tests) and percentages (Chi-Square test or Fisher’s test).

* P Value < .05.

† P Value < .001.

eGFR stands for estimated Glomerular Filtration Rate, SBP indicates systolic blood pressure, and DBP stands for diastolic blood pressure. The subscripts 24 h, d, and n represent 24 hours, daytime, and nighttime, respectively.

Comparison of the office BP, ABPM, and circadian rhythm according to the presence and absence of nocturnal hypotension

Table 1 shows that patients with NHP had 38 cases compared with those without 183 cases. The NHP group was more significant in older age (71.4 ±  7.1 vs. 68.7 ± 7.0, P = 0.048), had lower BMI (24.8 ± 3.1 vs. 26.0 ± 2.8, P = 0.021), and were less likely to combined hypertension (52.6% vs. 71.6%, P = 0.022) compared with the no NHP group. In addition, the nocturnal hypotension group had significantly lower diastolic BP at the clinic (80 (76–90) vs. 90(80–98), P = 0.006) and had lower ABPM in a 24 hour day, and night SBP and DBP than those of the no NHP group. There were no significant differences between the two groups in the duration of medication of sα1-RAs, fasting blood glucose, cholesterol, and eGFR (P > 0.05). No significant differences were in the proportion of diabetes, current drinkers, and smokers. Patients with nocturnal hypotension taken tamsulosin, doxazosin, and terazosin before sleep were 11(28.9%), 12(31.6%) and 13(39.5%) cases, repectively, compared with those without 85 (46.4%), 58(31.7%) and 40 (21.9%). There was significant correlation between sα1-RAs types and nocturnal hypotension (P = 0.046). Normal dippers in patients with nocturnal hypotension were 14 (36.8%) cases compared with those without 43 (23.5%). Abnormal circadian rhythm including nondippers, extreme-dippers, and reverse-dippers in the NHP group were 10 (26.3%), 14 (36.8%), and 0 (0%), while in the No NHP group were 98 (53.5%), 12 (6.6%), 30 (16.4%). No significant differences were found in the total abnormal circadian rhythm between the two groups 24(63.2%) vs. 140(76.5%) (P > 0.05).

Table 2 Univariate and multivariate logistic regression model for prediction of nocturnal hypotension.

Variable	Univariate model	Multivariate model	
	OR (95% CI)	P value	OR (95% CI)	P value	
Age, years	1.053 (1.005–1.104)	0.031*	1.064 (1.012–1.118)	0.015*	
Body Mass Index, kg/m2	0.854 (0.746–0.977)	0.022*			
Office DBP, mm Hg	0.961 (0.931–0.993)	0.016*			
sα1-RAs types (Tamsulosin as a reference)		0.053			
Doxazosin	1.599 (0.661–3.868)	0.298			
Terazosin	2.898 (1.221–6.875)	0.016*			
No Hypertension	2.267 (1.111–4.627)	0.024*	2.548 (1.211–5.359)	0.014*	
Notes.

OR indicates odds ratio, and CI indicates confidence interval.

* P Value < .05.

DBP stands for diastolic blood pressure.

In the univariate model, age, body mass index, office DBP, sα1-RAs types and no hypertension were separately included in logistic regression analysis as possible covariables.

In the multivariate model, factors such as age, length of taking sα1-RAs, body mass index, total cholesterol, fasting blood glucose, eGFR, office SBP, and DBP, current smoking, current drinking, diabetes, hypertension, sα1-RAs types and take other antihypertensive drugs at night were forced into a single model.

Factors associated with nocturnal hypotension

We calculated the odds ratio (OR) and 95% confidence interval (95% CI) using univariate and multiple logistic regression analysis to evaluate predictors of nocturnal hypotension (Table 2). The logistic binary regression analysis of the univariate model based on significantly different variables found that age (OR 1.053, 95% CI [1.005–1.104], P = 0.031), BMI (OR 0.854, 95% CI [0.746–0.977], P = 0.022), office DBP (OR 0.961, 95% CI [0.931–0.993], P = 0.016), no hypertension (OR 2.267, 95% CI [1.111–4.627], P = 0.024) and sα1-RAs types were predictive for nocturnal hypotension. The univariate regression model of sα1-RAs types using tamsulosin as a classification reference show that doxazosin (OR 1.599, 95% CI [0.661–3.868], P = 0.298), but terazosin (OR 2.898, 95% CI [1.221–6.875], P = 0.016), indicating the risk of nocturnal hypotension in patients taken terazosin was 2.898 times that of those taken tamsulosin. However, after multivariate adjustment incorporating sα1-RAs types as an additional adjustment factor, only age (OR1.064, 95% CI [1.012–1.118], P = 0.015) and no hypertension (OR 2.548, 95% CI [1.211–5.359], P = 0.014) remained significant associations with nocturnal hypotension. Each one-year increase in age was associated with 6.4% higher odds for the presence of nocturnal hypotension. No combination of hypertension was positively correlated with nocturnal hypotension independently. The risk of nocturnal hypotension in patients without hypertension was 2.548 times that of those with hypertension. The distribution of nocturnal hypotension among three age groups increased with age (P for trends < 0.05) and was significantly higher in the age group of ≥ 80 years than in the 60–69.99-year-old age group (30.8% vs. 11.7%, P < 0.05) (Fig. 1A). The occurrence of nocturnal hypotension was significantly higher in the no hypertension group than in the hypertension group (25.7% vs. 13.2%, P < 0.05) (Fig. 1B). There was no significant association between nocturnal hypotension and the duration of taking sα1-RAs, BMI, office SBP and DBP, fasting blood glucose, total cholesterol, diabetes, current alcohol consumption, and smoking.

Figure 1 Association of nocturnal hypotension distribution with age (A) and hypertension (B) in 221 older men with prostate hyperplasia given a bedtime dose of selective α1 adrenoceptor antagonists.

ROC curve analysis for age, BMI, and office DBP

Figure 2A showed a multivariate joint prediction ROC curve for nocturnal hypotension with age, body mass index, and office DBP. The area under the ROC Curve (AUC) of multi covariates was 0.731 (95% CI [0.64–0.82], P < 0.0001), and the negative and positive predictive power was 82.9% and 33.3%, respectively. The logistic plot of age was positive for nocturnal hypotension prediction (Fig. 2B). However, the logistic plot of office DBP (Fig. 2C) and BMI (Fig. 2D) were negative predictors for nocturnal hypotension. To determine the sensitivity and specificity of age, BMI, or office DBP for predicting the presence of nocturnal hypotension in older men with BPH after taking a bedtime dose of sα1-RAs, we performed ROC analysis (Fig. 3). The AUC of age used to predict nocturnal hypotension was 0.621 (95% CI [0.53–0.72], P < 0.05), the AUC of BMI was 0.607 (95% CI [0.50–0.72], P < 0.05), and the AUC of office DBP was 0.646 (95% CI [0.54–0.74], P < 0.05). Based on the ROC Curve Analysis, the optimal cut-off value for age was 68.08 years, and the sensitivity and specificity were 63.2% and 61.2%, respectively. The optimal cut-off value for BMI was 28.4, and the sensitivity and specificity were 18.9% and 83.4%, respectively. The optimal cut-off value for office DBP was 66.5 mmHg, and the sensitivity and specificity were 100% and 1.9%, respectively.

Figure 2 Multivariate joint prediction ROC curve (A) and logistic plots of age (B), office DBP (C), and BMI (D) for nocturnal hypotension.

Figure 3 ROC curve analysis for age, BMI, and DBPo (ROC, receiver operating characteristic; AUC, Area under the ROC curve; BMI indicates body mass index; DBPo stands for office diastolic blood pressure.

Discussion

Our study showed that nocturnal hypotension is prevalent in men aged 60 years and above with BPH treated with sα1-RAs before sleep. Nocturnal hypotension was positively associated with two independent risk factors: age and no hypertension. As a first-line treatment for BPH-related LUTS, sα1-RAs can also decrease systemic vascular resistance and reduce BP, especially in elderly patients (Jacobsen et al., 2008). Accumulated studies found that nocturnal hypotension episodes significantly increased in older adults with antihypertensive therapy (Cuspidi et al., 2019; Kantola et al., 2001). Our findings support recent observations (Albasri et al., 2021; Hanlon et al., 2021) on the potential harms of hypotension due to antihypertensive treatment and recommend rational BP management for older patients (Hanlon et al., 2021; Naschitz, 2018) to balance the benefits against the risks. Sα1-RAs are often taken before sleep, starting with a low dose and gradually increasing the dose based on their tolerance to avoid orthostatic hypotension. Severe hypotension (Bird et al., 2013) and secondary unwanted events (Jacobsen et al., 2008) were observed previously associated with α1-RA prescription no more than three months. Kario, Schwartz & Pickering (2000) observed a more pronounced reduction in nocturnal SBP in nondippers and risers with nighttime dosing of doxazosin. We investigated the effect on nighttime BP in older patients with BPH treated with conventional doses of sα1-RAs before sleep for at least three months with a steady drug blood concentration, which few studies did.

Hypotension by ABPM was defined mainly according to the predictive value of adverse cardiovascular events despite differences (Owens, Lyons & O’Brien, 2000; Kikuya et al., 2007; Ungar et al., 2009; Protogerou et al., 2007). A cohort study of a general population (Owens, Lyons & O’Brien, 2000) observed that patients with coronary artery disease with nighttime ABPM <90/50 mm Hg had a higher risk of ST-segment ischemia following the episode of hypotension. In a prospective population study (Kikuya et al., 2007) enrolled 5,682 participants (mean age 59.0 years), the approximate thresholds for optimal ABPM for nighttime amounted to 100/65 mm Hg considering the cardiovascular outcome. In a total of 3,090 person-years of the follow-up study, Ungar et al. (2009) found that the lowest DBP quartile (45–63 mm Hg) in the nighttime ABPM had the highest mortality risk in hypertensive patients >  60 years old. Protogerou et al. (2007) showed that a DBP ≤ 60 mm Hg value was independently associated with reduced survival in the elderly hypertensives treated. In our study concerning optimal organ perfusion in the elderly participants with BPH, including both hypertensives and non-hypertensives, we defined nocturnal hypotension by ABPM as the mean with nighttime SBP ≤ 100 mm Hg and/or DBP ≤ 60 mm Hg, close to most research standards above.

The prevalence of nocturnal hypotension by ABPM tended to agree with each other attributed to different thresholds of hypotension, different prescription drugs, different medication times, and different characteristics and compositions of the study populations. The sizeable Spanish cohort study (Divisón-Garrote et al., 2016) determined prevalence of hypotension was 8.2% with office BP and 3.9% with nighttime ABPM in 70,997 treated hypertensives. Nocturnal hypotension was defined as nighttime SBP < 90 and/or DBP < 50 mm Hg by ABPM. Accordingly, in those patients with nocturnal hypotension, 28% had only low SBP, 85.4% only low DBP, and 13.4% both, respectively. However, in the study (Matsumura et al., 2001) performed among 588 elderly patients (mean age 78 years), the episodes of nocturnal hypotension (SBP < 100 mmHg with 24-hour ABPM) were prevalent as high as 31%. In our study, the prevalence of nocturnal hypotension according to nighttime average SBP and/or DBP by ABPM was as considerably high as 17.2% after taking sα1-RAs before sleep in 221 elderly participants with or without hypertension. Among the patients with nocturnal hypotension, 23.7% had only low SBP, 31.6% had only low DBP, and 44.7% both by ABPM.

The related factors with nocturnal hypotension shown in our observation followed previous studies. In the Spanish ABPM Registry in hypertensives (Divisón-Garrote et al., 2016), independent factors associated with hypotension included age, female, BMI < 30 kg/m2, or previous ischemic heart disease. However, Scuteri et al. (2012) showed that the episodes of nocturnal hypotension were, unexpectedly, significantly more common in participants ≤ 75 years old than in the oldest group, even in hypertensives. Owens, Lyons & O’Brien (2000) reported that hypotensive individuals were of high proportion (49%) in a general population and more frequently found in women, specifically thin subjects with smaller muscle mass. Nevertheless, the study failed to show the difference between the sexes in the incidence of nocturnal hypotension. BMI decreased linearly with increasing age. Two Japanese cross-sectional studies (Masaki et al., 1997; Najjar, Scuteri & Lakatta, 2005) found a highly positive relationship between BMI and BP in aged participants. Low DBP was responsible for most hypotension in the aged (Divisón-Garrote et al., 2016), associated with more significant all-cause mortality (Ungar et al., 2009), and often occurred in treated hypertensives. Office DBP facilitates rapid detection of hypotension-prone elderly by physicians in clinical practice.

The current study provided some links with nocturnal hypotension for older participants but did not allow for speculation about potential mechanisms. Although tamsulosin, doxazosin, and terazosin are all clinically uroselective α1 adrenergic receptor antagonists, tamsulosin is selectivity for the α1A and α1D receptor subtypes and has less blood pressure lowering effect (Lowe, 2004; Lepor, Kazzazi & Djavan, 2012). Therefore, in the subgroup analyses based on the type of α1 receptor blockers the risk of nocturnal hypotensive events was lower with bedtime dosage of tamsulosin in contrast with the other two α1 antagonists in elderly with LUTS. Why do α1 blockers have a greater impact on blood pressure in elderly patients? Previous investigation (Rudner et al., 1999) provided a possible clue that there is an age-related predominant increase in vascular α1B receptors than α1A receptors. Hypotension in older age may also be related to the gradual degradation of physiological states (Najjar, Scuteri & Lakatta, 2005) as elastic artery dilation and stiffening, the renin-angiotensin-aldosterone system decline, and baroreflex sensitivity decrease. It may also be associated with a general deterioration in health, especially in frail patients (Rastas et al., 2006). Apart from antihypertensive treatment, various chronic diseases and comorbidities with related medications became causes of hypotension in elderly patients, such as chronic heart failure or sedative dosages for sleep disorders, et al. Some additional factors were secondary to sleep-related breathing disorders and elevated supine nasopharyngeal airway resistance leading to nocturnal hypotension (McGinty et al., 1988). The higher prevalence of nocturnal hypotension in elderly patients may suggest that it could be a multifactorial effect rather than a specific drug factor.

We acknowledge that our study has some limitations. First, as a cross-sectional study, causal relationships cannot be deduced from the results alone. Second, our findings may not be directly generalizable to other populations based on the specific participants: older men with BPH treated with a conventional dosage of sα1-RAs before sleep over three months, excluding those who could not tolerate α1-RAs and continued treatment course. Third, there was no data for comparison on subjects’ placebo-controlled or those who took drugs during another period. Furthermore, there is a need for information expansion on other comorbidities and related medications that may affect blood pressure in elderly patients during case collection. Finally, the information on patients’ sleeping and awake periods was far from complete; and an arbitrary blood pressure division between day and night was used.

Despite these limitations, our study has illustrated for the first time the association between nighttime dosing of sα1-RAs treatment and nocturnal hypotension in older men with BPH, including patients without hypertension underrepresented in most trials. The results need to be enhanced and expanded by multicenter prospective clinical studies focusing on the clinical categories: the context of one or several chronic diseases and comorbidities, in addition to advanced age and frailty degree, and further the related changes in target organ function with long-term follow-up which we are working.

Conclusions

Nocturnal hypotension was common and deserved particular attention when prescribing sα1-RAs to elderly patients, who constitute a large, growing, and vulnerable population. Assessing the physical condition, and related risk factors of nocturnal hypotension and chronic diseases in elderly patients, it is imperative to engage in thorough communication with them to select the plan that offers the greatest benefits and poses the lowest risks from the comprehensive treatment for BPH. Further genomic studies on variation expression levels of alpha(1)-AR subtypes in the prostate will lead to personalized administration and new strategies for clinical management. Recent years have seen the widespread use of novel drugs that are subtype-selective for the α1A adrenoceptor to treat BPH and new home blood pressure monitoring devices that minimize sleep disturbances (Kario, 2021). Future studies are needed to validate our research findings on nighttime blood pressure in elderly patients using these new drugs and devices. Our ongoing research will better characterize the clinical profile and predict the corresponding cardiovascular outcomes of abnormal nocturnal blood pressure in older adults.

Supplemental Information

Supplemental Information 1 Raw data

Supplemental Information 2 Flow diagram

Supplemental Information 3 Try protocol

Supplemental Information 4 STROBE checklist

The authors acknowledge the role of all participants recruited and the medical team without whom the study could not have been accomplished. Our manuscript has been edited partially by Grammarly.

Additional Information and Declarations

Competing Interests

Author Contributions

Human Ethics

Data Availability

The authors declare there are no competing interests.

Chao-Ting Chen conceived and designed the experiments, performed the experiments, analyzed the data, prepared figures and/or tables, authored or reviewed drafts of the article, and approved the final draft.

Shao-Jun Ma performed the experiments, analyzed the data, prepared figures and/or tables, and approved the final draft.

Hai-Ya Wang conceived and designed the experiments, authored or reviewed drafts of the article, and approved the final draft.

Hai-Jun Yao performed the experiments, prepared figures and/or tables, and approved the final draft.

The following information was supplied relating to ethical approvals (i.e., approving body and any reference numbers):

The Ethics Committee and Research Board of Shanghai Ninth People’s Hospital approved to carry out the study within its facilities (Ethical Application Ref: SH9H-2020-T25-2).

The following information was supplied regarding data availability:

The raw data is available in the Supplementary File.

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
