# Peer review of "A cross-sectional study on the effects of bedtime administration of selective α1 adrenoceptor antagonists on nocturnal blood pressure in elderly patients with benign prostate hyperplasia"

_PeerJ, doi:10.7717/peerj.19165_

## Round 0.1 · original submission · Major Revisions

Dear authors,

Manuscript titled "Effects on nocturnal blood pressure of a bedtime dose of selective α1 adrenoceptor antagonists in aged adults with benign prostate hyperplasia" that you submitted to PeerJ has been reviewed.

The reviewer(s) have suggested that some important points must be clarified and have requested substantial changes to be made in the manuscript. Therefore, I invite you to respond to the reviewer(s)' comments and revise your manuscript. The reviewer(s) comments are included at the end of this letter.

·

Basic reporting

This cross-sectional study investigates the effects of bedtime administration of selective α1-adrenoceptor antagonists (sα1-RAs) on nocturnal blood pressure in elderly patients with benign prostatic hyperplasia (BPH). The authors aim to assess the prevalence of nocturnal hypotension (NHP) in this population and identify associated risk factors. The study included 221 elderly male patients diagnosed with BPH, who had been treated with sα1-RAs. The findings suggest that nocturnal hypotension is relatively common in this population, with age and the absence of hypertension being significant risk factors.

The manuscript contains some grammatical errors and the writing style is overly complex in places, which might obscure the meaning. A more straightforward approach would improve readability. For instance, sentences like “Future investigations on the BPs during nighttime in elderly patients are warranted to validate our results obtained with novel drugs highly selective for the α1 adrenoceptor treating LUTS or with the newer home BP monitoring devices (Kario, 2021) with minimal sleep disturbance outcomes associated with outcomes.” are convoluted and could be simplified for better comprehension.

The title lacks clarity and specificity. It could be revised for better readability and to include the study type, e.g., "A Cross-Sectional Study on the Effects of Bedtime Administration of Selective α1 Adrenoceptor Antagonists on Nocturnal Blood Pressure in Elderly Patients with Benign Prostate Hyperplasia."

Experimental design

The findings of the manuscript are valuable, particularly in addressing benign prostatic hyperplasia (BPH), a common cause of urinary outflow obstruction in aging men that leads to lower urinary tract symptoms (LUTS). α1-Adrenoceptor (α1AR) antagonists (blockers) have become a cornerstone of LUTS treatment. However, these treatments are associated with complications, notably hypotensive episodes, especially in elderly patients. Despite the strengths of the study, the mentioned points need to be addressed in the manuscript.

Validity of the findings

Although the authors mentioned that they did not perform subgroup analyses based on the type or dose range of α1 blockers, as it was not their focus, this is a critical aspect to consider. Different α1 blockers and their dosages can lead to varying results and complications. For instance, tamsulosin, doxazosin, and terazosin have distinct pharmacological properties. Unlike doxazosin and terazosin, tamsulosin is more subtype-selective, targeting α1a/α1d receptors with 10 times greater affinity than α1b receptors. As a result, the risk of hypotensive events is higher with alfuzosin, prazosin, terazosin, and doxazosin, which have the lowest uroselectivity. In contrast, tamsulosin has been shown to have a lesser impact on blood pressure in both hypertensive and normotensive patients with LUTS. Therefore, incorporating these factors into the statistical analysis, such as through group comparisons or logistic regression, is essential.

Hypotensive episodes in patients using α1 blockers are more common in older individuals, likely due to an age-related increase in vascular α1B receptors. As a result, α1 blockers have a greater impact on blood pressure in elderly patients, where vascular α1B receptors become more predominant than α1A receptors. However, this point is not thoroughly discussed in the manuscript.

The observation that non-hypertensive patients are more likely to experience nocturnal hypotension is interesting, but the proper discussion behind this finding are not explored in depth. Moreover, The suggestion to avoid α1 blockers in certain elderly populations is crucial, but the discussion should include a more detailed consideration of alternative therapies and management strategies.

Reviewer 2 ·

Basic reporting

Summary
This observational study assessed the association of a bedtime dose of selective α1 adrenoceptor antagonist and the presence of nocturnal hypotension in elderly patients with benign prostate hyperplasia (BPH) (n=221, mean age 69 years). In the total group, 38 patients showed nocturnal hypotension. This study demonstrated that age and no hypertension were associated with the presence of nocturnal hypotension.

Comments
・The authors should describe their hypothesis of this study in the Introduction section.

・It is recommended that uncommon abbreviations such as nocturnal hypotension (NHP) not be used.

・It would be better to include the number (percentage) of patients who took antihypertensive drugs after dinner or before bedtime in Table 1.

・The adjustment factors in the legend of Table 2 is different from those described in the main text. This is an important point, so the authors should revise this point clearly.

Experimental design

This study was well-designed and the methods section was well-written.

Validity of the findings

This reviewer thought that the validity of this study was correct.

Additional comments

・This reviewer would like to ask the authors that why they assessed the dipping pattern in this study. They should clarify the purpose of them.

・What was the rationale of the definition of nocturnal hypotension of this study? This point should be described in the methods section.

---

## Round 0.2 · accepted · Accept

Dear Author,

Congratulations! After your diligent work addressing the reviewers' comments, I am pleased to inform you that your manuscript has been accepted for publication in PeerJ. This version is more concise and formal, enhancing clarity and flow.

·

Basic reporting

Concerns have been resolved.

Experimental design

Concerns have been resolved.

Validity of the findings

Concerns have been resolved.

Additional comments

Concerns have been resolved.

Reviewer 2 ·

Basic reporting

no comment

Experimental design

no comment

Validity of the findings

no comment

Additional comments

Issues were addressed in the present revision. I have no further comments.